# Polio vaccination and marginal effects among children aged 12–23 months in Lesotho using 2024 LDHS data: A multilevel analysis approach

Kassahun Animut Metkie[1]*, Yordanos Sisay Asgedom[2],
Seblewongel Gebretsadik Sertsewold[3], Mesfin Abebe[4],
Amanuel Yosef Gebrekidan[5], Tsion Mulat Tebeje[6]

**1** Department of Statistics, College of Natural and Computational Science, Dilla University, Dilla, Ethiopia, **2** Department of Epidemiology and Biostatistics, College of Health Sciences and Medicine, Wolaita Sodo University, Wolaita Sodo, Ethiopia, **3** Asrat Woldeyes Health Science campus, Debrebirhan University, Debrebirhan, Ethiopia, **4** Department of Midwifery, College of Health Sciences and Medicine, Dilla University, Dilla, Ethiopia, **5** School of Public Health, College of Health Sciences and Medicine, Wolaita Sodo University, Wolaita Sodo, Ethiopia, **6** School of Public health, College of Health Sciences and Medicine, Dilla University, Dilla, Ethiopia

* kaas.ww@gmail.com

## Abstract

### Background

Infectious diseases that have affected countless newborns, children, and adults can be avoided with polio vaccination. The WHO and Lesotho's health strategy placed a strong emphasis on vaccination as a means of preventing and controlling infectious diseases. The current study's goal was to indicate marginal effects and identify the determinants of polio vaccination status in children between the ages of 12 and 23 months.

### Methods

The polio vaccination status in Lesotho was evaluated using LDHS data collected from 490 children aged 12–23 months. The prevalence of polio vaccination was presented through frequencies and percentages. The heterogeneity in child polio vaccination status across clusters was determined using the ICC value and the chi-square test. The data was analyzed using a multilevel ordinal model, and model selection was based on IC values.

### Results

The descriptive findings of the study revealed that approximately 29.6% of children were not fully vaccinated against polio. The use of multilevel analysis was deemed appropriate based on a significant chi-square result (p-value = 0.033). Clusters accounted for approximately 15.9% (ICC = 15.88) of the overall variation in polio

**Data availability statement:** The data used in this study is available freely online from DHS programs website i.e. (https://www.dhsprogram.com/data).

**Funding:** The author(s) received no specific funding for this work.

**Competing interests:** The authors have declared that no competing interests exist.

**Abbreviations:** OPV, oral polio virus; IPV, inactivated polio virus; WHO, world health organization; LDHS, Lesotho demographic and health survey; ICC, intra-class correlation; AIC, Akaike information criteria; BIC, Baysian information criteria; ME, marginal effect; PO, proportional odds; ANC, antenatal care.

vaccination status among children. The final adjusted random intercept PO model indicated a significant relationship (p-value < 0.05) between children's polio vaccination status and factors such as district type, health facility visits in the past 12 months, number of ANC visits, place of delivery, and family size. The marginal effects of these key variables were also significant for each category of polio vaccination status.

## Conclusions

In Lesotho and around the world, polio vaccination is a crucial health measure to enhance children's development. Approximately 29.6% of the children in the study were not fully vaccinated. These rates differed by district and the number of ANC visits the moms had. Treatments and safeguards for children's health should be more focused on enhancing medical facilities and services and encouraging ANC visits to receive care.

## Introduction

One of the most cost-effective and efficient public health interventions to prevent vaccine-preventable diseases is childhood immunization. It saves between 3.5 and 5 million lives worldwide [1]. Polio is a crippling and deadly disease that can be safely and effectively prevented by a vaccine. Since 1988, global efforts to immunize children against the poliovirus have resulted in a 99.9% reduction in the wild poliovirus [2].

The World Health Organization (WHO) estimates that between 2018 and 2021, about one million children under five will not receive the polio vaccine [3]. There are 185 non-endemic and 52 epidemic polio countries in the world [4]. WHO has proposed a combined Oral polio vaccine (OPV) and Inactivated polio vaccine (IPV) program to achieve the global strategic goal of polio eradication, a programmatic priority for global public health [5]. Both vaccines are used in the global poliomyelitis eradication effort to prevent the spread of the disease. Poliomyelitis is being eradicated in part by inactivated (IPV) and oral (OPV) poliovirus vaccines. Although the oral poliovirus vaccine is quite effective, outbreaks of circulated vaccine derived poliovirus (cVDPV) or vaccine-associated paralytic poliomyelitis (VAPP) are rare. High immunization coverage is needed to prevent outbreaks and stop the spread of polio [6].

The wild poliovirus has been stopped in Africa, but the main problems remain circulating vaccine-derived poliovirus due to low vaccination rates, places that are difficult to access, and a lack of planning and money [7]. The African Region's routine vaccination program and health system were significantly interrupted by the COVID-19 pandemic in 2020 and 2021, which resulted in a sharp drop in vaccine coverage. Stable communities and operational health facilities had importance for routine immunization programs [8].

In 2022, 71% and 73% of children in the African Region are estimated to have received OPV3 and IPV1, respectively [8]. OPV3 coverage declined from 77% in the year 2019 to 73% in the year 2020, 72% in the year 2021, and 71% in the year 2022 [8]. The goal of the 2023 polio campaigns was to vaccinate 21 million children in Cameroon, Chad, Niger, and Mozambique, all considered very high risk. The legacy of the polio initiative, such as alleviating suffering and building a more robust immunization infrastructure, is part of the success of the African polio eradication effort [9].

Lesotho has been certified polio-free since 2005, when the African Regional Certification Commission (ARCC) approved the country's documentation [10]. However, Lesotho remains at risk of polio importation due to cases in other regions of Africa. Lesotho's polio immunization program consists of one dose of inactivated polio vaccine (IPV) at 14 weeks and four doses of bivalent oral polio vaccine at birth, 6, 10, and 14 weeks. One of the first countries in Africa to include the second dose of IPV in its routine immunization schedule was Lesotho [11].

In seven of ten districts in Lesotho, around 20% of children did not receive vaccinations in 2021 and 2022. The third dose of the polio vaccine was administered to 86.75% of children in Lesotho as of 2023 [12]. Over the past five years, Lesotho has achieved non-polio acute flaccid paralysis rate of over two cases per 100,000 people aged fewer than fifteen [11]. Inadequate health workers understanding of polio, quiet reporting in certain locations and a lack of community and environmental surveillance systems are some of the obstacles to acute flaccid paralysis surveillance. Children are still at risk of poliovirus infection due to stakeouts at the service delivery level [11].

Lesotho has a small number of studies on basic factors and child vaccination coverage [11]. However, the purpose of this study was to evaluate the level of polio vaccination variability between clusters and the marginal impacts of each explanatory variable on polio vaccination categories among Lesotho children between the ages of 12 and 23 months.

## Methods

### Study data source, population and study design

Childhood polio vaccination and marginal effects were examined using a secondary dataset from the Lesotho Demographic and Health Survey (2023–2024 LDHS), which was available on the DHS programme website upon formal request. The LDHS was a community-based cross-sectional study conducted in Lesotho from 27 November 2023 to 29 February 2024. Administratively, Lesotho is divided into ten districts (Butha-buthe, Leribe, Berea, Maseru, Mafeteng, Mohale's Hoek, Quthing, Qacha's Nek, Mokhotlong, and Thaba-tseka); each district is further divided into constituencies and each constituency into local councils [13].

Households in the LDHS sample were selected and stratified separately in two stages. Based on the 2016 Lesotho population and housing census, 29 sampling strata were formed after each district was classified into urban, peri-urban, and rural sectors. In the first sampling stage, 400 EAs in each sampling stratum were selected independently with probability based on EA size. In the second round of selection, 25 households per cluster (EA) were systematically selected with equal probability from the newly generated list of households [13]. The study population of this study was all children aged between 12 and 23 months in the selected enumeration areas (EAs) or clusters.

From all these selected households, the polio vaccination status of 529 children aged between 12 and 23 months were collected in 2024 LDHS data and we extracted the sample from this data set (Fig 1).

### Data extraction method

A permission letter was obtained to access and use 2023/2024 LDHS data from the Demographic and Health Survey (DHS) program website (http://www.dhsprogram.com). Data cleaning, data extraction, re-coding of variables, variable computations, and potentially associated variable selections were all carried out in accordance with relevant literature.

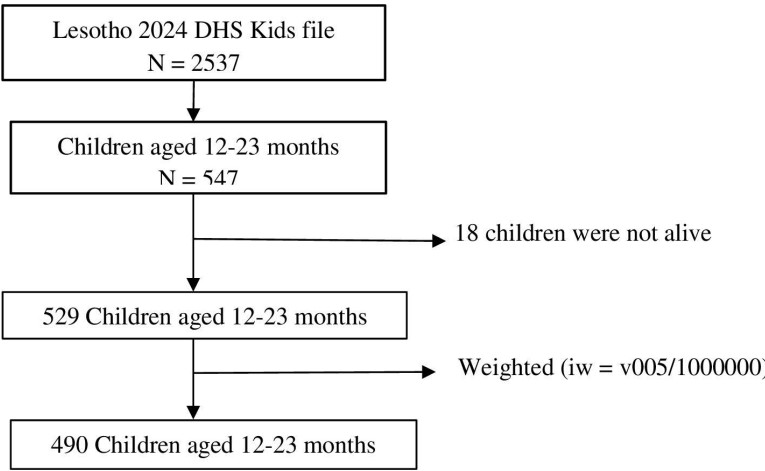

**Fig 1. Flowchart of the study sample.**

## Inclusion-exclusion criteria

In contrast to children who had missing or unknown polio vaccination status, all 12–23 month-old Lesotho children with known polio vaccination status were taken into consideration for this study.

## Ethics approval

There was no need for ethics approval because this study used secondary DHS data that was collected in accordance with international ethical principles and guidelines.

## Study variable measurements

**Dependent variable.** The outcome variable of the study was the polio vaccination status of Lesotho children aged 12–23 months. This was determined by counting the number of doses of polio vaccine a child had received. Children who were fully immunized against polio received all four recommended doses (zero, one, two, and three doses). A child who had received some of the required doses of polio vaccine was considered incomplete vaccinated, while a child who had not received any of the recommended doses was considered not-vaccinated [14].

**Independent variables.** The independent variables for the child polio vaccination status were chosen based on theoretical ideas and relevant literature on child immunization [1,15]. In this study demographic, socio-economic and health related variable was considered. These independent variables are age mothers, marital status, occupation type, residence type, number of antenatal care visits, mother education level, family size, wealth index, parity, number of births in last five years, visit health facility in last 12 months, and place of delivery.

## Statistical analysis

The extracted polio vaccination data were examined for completeness and weighted to account for the variation in interview and selection chance among cases in the chosen sample. STATA (15) was used to analyze the data, and both descriptive (frequency, percentage, and charts) and inferential statistical techniques were used to display the results. Chi-square analysis was used to examine the relationship between explanatory factors and polio vaccination status and used to select potential candidate variables with a p-value of 0.15 [16]. The multilevel ordinal logistic regression model was used to identify independent variables that were statistically significant [16].

**Multilevel ordinal regression model.** Logistic regression is a technique used to analyze the presence or absence of a specific response variable in relation to potential explanatory factors. For this study, logistic regression represents the relationship between explanatory variables and the polio vaccination status level as the logarithm of the odds (logistic transformation). Ordinal logistic regression was employed for polio vaccination status with ordinal effects, as it utilizes the cumulative distribution of polio vaccination status values. The parameters it estimates for each association reflect the overall trend across the ordinal levels of the child polio vaccination status [17].

The parallel line assumption asserts that every independent variable exerts the same influence across all cumulative categories of the polio vaccination levels, meaning the parameters remain consistent across different polio vaccination level categories [16]. The likelihood ratio test, Wald Chi-Square test, score test, and other related methods are utilized to assess the parallel lines assumption [18]. Equality 1 tests the hypothesis under the parallel line assumption to determine whether the $\beta_k$ coefficients of the independent variable are equal across polio vaccination level categories [19].

$$H_0 = \beta_{1j} = \beta_{2j} = \ldots = \beta_{(k-1)j} = \beta \qquad j = 1, 2, \ldots, J \tag{1}$$

The fundamental types of ordinal logistic regression models employed when this parallel line assumption was violated are the partial proportional odds model, the continuation ratio model, and the generalized ordered logit model [20].

The lack of independence between child polio vaccination status and clusters of data is explained by a hierarchical or multilevel logistic regression model. By considering the polio vaccination variability associated with each level (children and clusters) of the hierarchy, statistical models known as hierarchical models are used to investigate these nested causes of variability [21].

**Two level model.** The basic multilevel ordinal model based on generalized linear models uses the cumulative probabilities of response categories as the dependent variables changed [22]. The two-level proportional odds model with a logit link is:

$$logit(\pi_{ir(j)}) = \log\left(\frac{\pi_{ir(j)}}{1 - \pi_{ir(j)}}\right) = \alpha_j - (\beta x_{ir} + U_{0r}), \ j = 1, 2, \ldots, J-1 \tag{2}$$

The variability of the proportions between the clusters was analyzed using the chi-square statistic in order to be able to use the multilevel analysis. The intra-class correlation quantifies the degree of similarity between children vaccination in the same clusters or randomly selected children from the same cluster [23]. The model goodness of fit test for multilevel model is based on the Akaki information and Bayesian information criteria in addition to the deviance values of the models [24].

**Parameter estimation for multilevel model.** The common methods in parameter estimation for this study are based on the likelihoods. Marginal Quasi-likelihood and Penalized Quasi-likelihood are approximate methods [25].

## Marginal effects

The instantaneous change in the response probability that will result from a unit change in covariates is measured by marginal effects for continuous variables. The difference in the expected probabilities for cases in one category compared to the reference category was demonstrated by the marginal effects for the logit models of the categorical variable (polio vaccination status) with more than two potential values.

Marginal effects are used to quantify discrete change in categorical independent variables. That is, how do the predicted probabilities of polio vaccination level change as the categorical independent variable shifts from one category to the other? When all other variables are held equal, marginal effects can be a useful tool for summarizing the relationship between changes in the polio vaccination status and discrete changes in a categorical variable [26,27].

## Results

### Demographic health related and other characteristics

The research utilized data from the 2023/2024 LDHS concerning the vaccination status for polio among children. The study focused on 490 weighted number of children aged 12–23 months, with 27 (5.5%) of them not vaccinated, 146 (29.7%) either not vaccinated or incompletely vaccinated, and 344 (70.4%) fully vaccinated (Table 1).

The distribution of polio vaccination status by demographic, health-related, and other variables among children aged 12–23 months is shown in Table 2. Approximately 292 (59.6%) of the 490 children who participated in the survey were from rural areas. The percentages of not vaccinated, incompletely vaccinated, and fully vaccinated children from the rural areas were 6.1%, 21.5%, and 72.4%, respectively. In the Maseru district, the percentages of children aged 12–23 months who were not polio vaccinated, partially vaccinated, and fully vaccinated were 11.2%, 29.2%, and 59.6%, respectively. However, Butha-Buthe, Quthing, and Mokhotlong had lower percentages of children who were not vaccinated than other districts (Table 2).

The percentage of unvaccinated and partially vaccinated children was greater among those whose mothers fell within the age ranges of 35–49 and 25–34. The occurrence of both unvaccinated and partially vaccinated children against polio was higher (6% and 25.3%, respectively) among those whose mothers had only primary education or none at all than among children whose mothers attained secondary or above education level. Children from households with a higher number of members (≥ 6) showed an increased rate of incomplete polio vaccination status (Table 2).

Among children whose mothers did not work, the percentages of unvaccinated and partially vaccinated children against polio were 5.6% and 21.1%, respectively. Children from families with the lowest wealth index had rates of incomplete and non-vaccinated status of 22.3% and 4.4%, respectively. The more antenatal care (ANC) visits there were, the higher the prevalence of adequate vaccination status. The prevalence of incomplete and non-vaccinated children increased in tandem with the total number of children born. The percentages of unvaccinated and incompletely vaccinated children whose mothers were exposed to the media were 7.0% and 24.8%, respectively (Table 2).

The prevalence of complete vaccination among children of mothers who visited health facilities during the past 12 months was around 72.4%, while children of mothers who delivered at home had higher rates of incomplete (26.7%) and non-vaccination (13.4%) against polio. About 71.2% of children born to mothers who gave birth in a medical facility were completely vaccinated against polio. Children in households with significant challenges in distance to health centers had a higher proportion of both unvaccinated (9.4%) and partially vaccinated (26.2%). The percentages of children who were unvaccinated (5.5%) or partially vaccinated (24.2%) against polio were higher among families without health insurance. Approximately 76.9% of children born from mothers who had health insurance were fully vaccinated (Table 2).

### Multilevel ordinal logistic regression model

In this study, a significant chi-square test result ($\mathcal{X}^2 = 3.38$, and p = 0.033) confirmed the presence of heterogeneity in polio vaccination across clusters in Lesotho. The intra-class correlation coefficient result (ICC = 0.1588) also explained that the level two units (clusters) account for or explain approximately 15.88% of the overall variability of polio vaccination status.

**Table 1. Proportions of polio vaccination status among children aged between 12-23 months, LDHS 2023/2024 (weighted *n* = 490).**

| Vaccination status | Freq. | percent | Com. Percent |
|---|---|---|---|
| Not-vaccinated | 27 | 5.5 | 5.5 |
| Incomplete vaccinated | 119 | 24.2 | 29.7 |
| Fully vaccinated | 344 | 70.4 | 100 |

**Table 2. Demographic/health related and other characteristics of children's polio vaccination status LDHS 2023/2024 (weighted *n* = 490).**

| Variables | Not vaccinated Count (%) | Incomplete vaccinated Count (%) | Full vaccinated Count (%) | Total |
|---|---|---|---|---|
| **Age of mothers** | | | | |
| 15-24 | 10(4.8) | 50(23.3) | 153(71.9) | 213 |
| 25-34 | 10(5.2) | 47(24.2) | 135(70.6) | 192 |
| 35-49 | 7(7.8) | 22(26.4) | 56(65.8) | 85 |
| **Residence** | | | | |
| Urban | 9(4.5) | 56(28.2) | 133(67.3) | 198 |
| Rural | 18(6.1) | 62(21.5) | 212(72.4) | 292 |
| **Educ. Level** | | | | |
| Primary/below | 8(6.0) | 36(25.3) | 98(68.7) | 142 |
| Sec/above | 18(5.2) | 83(23.7) | 247(71.1) | 348 |
| **Wealth index** | | | | |
| Poorest | 5(4.4) | 24(22.3) | 79(73.3) | 108 |
| Poorer | 4(4.3) | 19(20.7) | 70(75.0) | 93 |
| Middle | 10(10.0) | 20(21.6) | 65(68.4) | 95 |
| Richer | 5(5.3) | 29(28.7) | 67(66.0) | 101 |
| Richest | 3(3.3) | 26(27.6) | 64(69.1) | 93 |
| **Antenatal visit** | | | | |
| <4 | 4(6.0) | 22(30.9) | 45(63.1) | 71 |
| 4-7 | 13(4.8) | 70(24.6) | 199(70.6) | 282 |
| ≥8 | 9(7.0) | 26(20.2) | 95(72.9) | 130 |
| **Occupation** | | | | |
| Not working | 16(5.2) | 65(21.1) | 225(73.8) | 306 |
| Working | 11(6.0) | 54(29.4) | 119(64.6) | 184 |
| **Parity** | | | | |
| one | 14(6.0) | 52(23.2) | 160(70.8) | 226 |
| 2-3 | 8(3.8) | 48(23.7) | 148(72.5) | 204 |
| ≥4 | 5(9.1) | 18(29.3) | 37(61.6) | 60 |
| **Media exposure** | | | | |
| No | 7(3.3) | 47(23.3) | 149(73.4) | 203 |
| Yes | 20(7.0) | 71(24.8) | 196(68.2) | 287 |
| **Family size** | | | | |
| 1-5 | 19(6.6) | 66(23.2) | 200(70.2) | 285 |
| ≥6 | 9(3.9) | 52(25.6) | 144(70.5) | 205 |
| **Visit HF 12m** | | | | |
| No | 8(9.0) | 27(29.6) | 56(61.4) | 91 |
| Yes | 18(4.7) | 92(23.0) | 289(72.4) | 399 |
| **Marital status** | | | | |
| Married/live partn | 7(7.1) | 19(22.4) | 61(70.5) | 87 |
| Unmarried | 20(5.1) | 99(24.6) | 283(70.3) | 403 |
| **Delivery place** | | | | |
| Home | 5(13.4) | 10(26.7) | 22(59.8) | 36 |
| Health institution | 22(4.8) | 109(24.0) | 323(71.2) | 454 |
| **Literacy status** | | | | |
| Illiterate | 0(0.0) | 1(14.8) | 7(85.2) | 8 |
| Literate | 27(5.5) | 117(24.4) | 338(70.1) | 482 |

*(Continued)*

**Table 2.** (Continued)

| Variables | Not vaccinated Count (%) | Incomplete vaccinated Count (%) | Full vaccinated Count (%) | Total |
|---|---|---|---|---|
| **Distance to HF** | | | | |
| Big problem | 12(9.4) | 34(26.2) | 82(64.4) | 128 |
| Not big problem | 15(4.1) | 85(23.4) | 262(72.5) | 362 |
| **Health Insurance** | | | | |
| No | 27(5.5) | 117(24.2) | 338(70.2) | 482 |
| Yes | 0(0.0) | 2(23.1) | 6(76.9) | 8 |
| **District** | | | | |
| Butha-buthe | 0(0.0) | 8(25.5) | 24(74.5) | 32 |
| Leribe | 3(3.5) | 14(17.9) | 61(78.6) | 77 |
| Berea | 2(2.6) | 11(17.5) | 49(79.9) | 62 |
| Maseru | 19(11.2) | 48(29.2) | 99(59.6) | 166 |
| Mafeteng | 1(3.4) | 5(20.5) | 18(76.1) | 24 |
| Mohale's hoek | 1(1.5) | 7(27.0) | 19(71.4) | 27 |
| Quthing | 0(0.0) | 4(26.4) | 12(73.5) | 16 |
| Qacha's nek | 1(4.9) | 4(27.1) | 11(68.0) | 17 |
| Mokhotlong | 1(0.8) | 6(25.8) | 17(73.4) | 24 |
| Thaba-tseka | 2(3.5) | 10(22.2) | 33(74.3) | 45 |

An insignificant Brant test result (p-value = 0.742) supported the proportional odds assumption, and the coefficient values in the dichotomization of polio vaccination status did not change. As a result the multilevel proportional odds models with null, random intercepts, and random coefficients were taken into consideration. The final model used to determine significant predictors of polio vaccination status and parameter estimates at a 5% significance level was the random intercept POM, as it had the smallest IC values (Table 3).

**Results of random intercept proportional odds model.** The contrasting panels shown in Table 4 compared non-vaccinated children to those who were either incompletely or fully vaccinated, as well as comparing both non-vaccinated and incompletely vaccinated children to those fully vaccinated. In the random intercept proportional odds model, predictors were added, resulting in a reduction of the variance component in the random effect from 1.99 in the null model to 0.379. This decrease in variance was attributed to the inclusion of fixed predictors, demonstrating a notable variation in child polio vaccination status among different clusters in Lesotho.

All of the independent variables in the model had constant coefficients for each of the response categories. The results Wald chi-square test of the random intercept PO model showed that the following factors were substantially related to the child polio vaccination status: district (p-value = 0.038), number of ANC visits (p-value = 0.011), number of health facility visits within the previous 12 months (p-value = 0.007), delivery place (p-value = 0.012), and family size (p-value = 0.031) (Table 4).

According to the fitted model, children from the Maseru district had a 2.7 (OR = 2.67, p-value = 0.032) times higher chance of having a better polio vaccination status than children from the Butha-Buthe district. Similarly, the fitted model showed that a child in the Qacha's Nek area was approximately 27% (OR = 0.72, p-value = 0.041) less likely to be partially or totally vaccinated compared to not being polio immunized than a child in the Butha-Buthe district. In Lesotho, children aged 12–23 months who came from families of six or more were approximately 2.3 (OR = 0.44, p-value = 0.011) times more likely to report having a worse polio vaccination status (none or incomplete vaccination) than children from families of less than five when all other model variables were held constant (Table 4).

 

**Table 3. AIC and BIC values for multilevel proportional models.**

| Model | Observations | AIC | BIC |
|---|---|---|---|
| Null model | 519 | 750 | 826 |
| Random intercept model | 519 | 712 | 725 |
| Random coefficient model | 519 | 751 | 832 |

**Table 4. Results of Parameter estimates of random intercept PO model.**

| Independent variables | Categories | OR | P-value | 95% CI | Wald test p-value |
|---|---|---|---|---|---|
| Districts (Ref: Butha-buthe) | Leribe | 1.20 | 0.708 | (0.46, 3.13) | 0.038 |
| | Berea | 1.36 | 0.533 | (0.52, 3.53) | |
| | Maseru | 2.67 | 0.032 | (1.12, 6.48) | |
| | Mafeteng | 1.22 | 0.721 | (0.40, 3.74) | |
| | Mohale's hoek | 0.89 | 0.816 | (0.35, 2.29) | |
| | Quthing | 1.14 | 0.811 | (0.41, 3.13) | |
| | Qacha's nek | 0.72 | 0.041 | (0.28, 0.53) | |
| | Mokhotlong | 0.89 | 0.09 | (0.35, 2.27) | |
| | Thaba-tseka | 0.97 | 0.940 | (0.41, 2.32) | |
| Visit health facility in last 12 months (Ref: No) | Yes | 2.39 | 0.023 | (1.78, 4.71) | 0.007 |
| Number of ANC visits (Ref: <4) | 4-7 | 1.38 | 0.260 | (0.79, 2.41) | 0.011 |
| | More/equal to 8 | 2.20 | 0.002 | (1.10, 4.26) | |
| Mother occupation (Ref: not working) | Working | 0.79 | 0.344 | (0.49, 1.28) | 0.122 |
| Delivery place (Ref: home) | Health institution | 2.64 | 0.006 | (1.52, 4.01) | 0.012 |
| Family size (Ref: 1–5) | More/equal to 6 | 0.44 | 0.011 | (0.22, 0.72) | 0.031 |
| | Constant-1 | -2.58 | | (-3.74, -1.42) | |
| | Constatnt-2 | -0.25 | | (-1.07, 1.07) | |

**Cluster level variance=0.379\*\***

The final model result showed that children of Lesotho mothers who visited health facilities within the previous 12 months were 2.4 (OR=2.39, p-value=0.023) times more likely to be in the highest vaccinated category than children whose mothers did not visit health facilities during the same time period. Similarly, when all other factors were held constant, children aged 12–23 months who were born to mothers who had more than or equal to eight ANC visits were 2.2 (OR=2.20, p-value=0.002) times more likely to report a better polio vaccination status (incomplete and fully or fully vaccinated) than children whose mothers had fewer than four ANC visits. A child born to Lesotho mothers who gave birth in a health facility had a 2.6 (OR=2.64, p-value=0.006) higher chance of receiving either a full and partial polio vaccination, or the reverse of non-vaccination status, when all other factors were held constant (Table 4).

## Results of marginal effects

The predictor variables, such as district, number of ANC visits, place of delivery, family size, and attendance at a health facility in the last 12 months, had significant marginal effects on the child's polio vaccination status (Table 5).

The results of the marginal effect of predictors showed that, in comparison to children from the Butha-buthe district in Lesotho, the probability of incomplete polio vaccination increased by 16 (ME=0.16, p-value=0.023) percentage points, while the likelihood of children aged 12–23 months in the Maseru district not being vaccinated decreased by about 25 (ME=-0.25, p-value=0.007) percentage points. Similarly, children in the Maseru district had a roughly 21 (ME=0.21,

p-value = 0.024) percentage point higher chance of being fully vaccinated than children in the Butha-Buthe district from Lesotho (Table 5).

According to the marginal effects, the probability that children in Lesotho would be partially and completely vaccinated increased by approximately 5 (ME = 0.044, p-value = 0.018) and 16 (ME = 0.16, p-value = 0.031) percentage points, respectively, when the mothers visited health facilities during the previous 12 months. Compared to children of mothers who had fewer than four ANC visits, the likelihood that their children would be fully polio vaccinated increased by 7 (ME = 0.07, p-value = 0.027) percentage points when the mothers' ANC visits were between 4 and 7. Additionally, the results demonstrated that children of mothers who visited eight or more times were approximately 12 percentage points less likely to be incompletely vaccinated (ME = -0.12, p-value = 0.025) than children of mothers who visited less than four times. In a similar vein, children's chances of receiving all recommended vaccinations improved by 15 percentage points when the mother's ANC visits exceeded 8 times (ME = 0.146, p-value = 0.026).

The likelihood of being unvaccinated and incompletely vaccinated decreased by almost 13 (ME = -0.127, p-value = 0.015) and 11 (ME = -0.110, p-value = 0.007) percentage points, respectively, for children whose mothers gave birth at medical facilities as opposed to those whose mothers gave birth at home. In contrast, there was a 14 percentage point increase in the likelihood of being completely vaccinated among children from mothers who were delivered in a health facility (ME = 0.139, p-value = 0.008). When all other factors were held constant, the probability of being completely vaccinated among children from families with a size of six or more was 21 percentage points lower than that of children from families with a size of less than five (ME = -0.21, p-value = 0.012) (Table 5).

## Discussion

Poliomyelitis, sometimes known as polio, is a highly contagious disease that can paralyze or kill children. Polio vaccines are used to prevent this disease. Using the number of doses administered as an ordinal outcome, the current study examined the polio vaccination status of children between the ages of 12 and 23 months. Children's polio vaccination status revealed that roughly 70.4% of them had received all recommended doses, whereas 5.5% had not. AIC and BIC values

**Table 5. Marginal effects (ME) of predictors on polio vaccination status, LDHS 2023/2024.**

| Predictors | | Not vaccinated | | Incomplete vaccinated | | Fully vaccinated | |
|---|---|---|---|---|---|---|---|
| | | ME1 | P-value | ME2 | P-value | ME3 | P-value |
| **District** **(Ref: Butha-buthe)** | Leribe | -0.04 | 0.708 | -0.03 | 0.708 | 0.04 | 0.708 |
| | Berea | -0.07 | 0.537 | -0.05 | 0.532 | 0.05 | 0.532 |
| | Maseru | -0.25 | 0.007 | 0.16 | 0.023 | 0.21 | 0.024 |
| | Mafeteng | -0.005 | 0.725 | -0.03 | 0.728 | 0.04 | 0.727 |
| | Mohale'shoek | -0.03 | 0.817 | 0.02 | 0.816 | -0.03 | 0.816 |
| | Quthing | -0.04 | 0.809 | -0.02 | 0.810 | 0.02 | 0.810 |
| | Qacha's nek | 0.01 | 0.515 | 0.05 | 0.040 | -0.07 | 0.044 |
| | Mokhotlong | 0.04 | 0.811 | 0.03 | 0.810 | -0.03 | 0.810 |
| | Thaba-tseka | 0.008 | 0.949 | 0.04 | 0.949 | -0.05 | 0.949 |
| **Visit health facility in last 12 months** **(Ref: No)** | Yes | -0.08 | 0.375 | 0.044 | 0.018 | 0.16 | 0.031 |
| **Number of ANC visits** **(Ref: < 4)** | 4-7 | -0.02 | 0.308 | -0.06 | 0.265 | 0.07 | 0.027 |
| | More/equal to 8 | -0.03 | 0.059 | -0.12 | 0.025 | 0.146 | 0.026 |
| **Mother occupation (Ref: not working)** | Working | 0.07 | 0.372 | 0.04 | 0.353 | -0.05 | 0.354 |
| **Delivery place** **(Ref: home)** | Health institutions | -0.127 | 0.015 | -0.110 | 0.007 | 0.139 | 0.008 |
| **Family size (Ref: 1–5)** | More/equal to 6 | 0.03 | 0.614 | 0.02 | 0.613 | -0.21 | 0.012 |

were used to compare the models for the multilevel PO models. The model with the smallest AC values was the random intercept POM. For the final model, parameter estimates for the significant predictors were shown and interpreted at a 5% significance level. Child polio vaccination status was significantly affected by the independent variables of district, number of ANC visits, visits to health institutions in the last 12 months, place of delivery, and family size.

This study discovered a relationship between a child's district in Lesotho and their polio vaccination status. Compared to Butha-Buthe district children, Maseru district children had a higher polio vaccination level. This result was supported by studies conducted in Ethiopia [28] (non-immunization rates were higher in Afar and Somalia), India [29], and Pakistan [1]. The plausible explanation for this outcome could be that vaccine distribution varies across districts due to geographical challenges (remote or hard-to-reach areas) in Lesotho or may be due to income level differences for limited access to polio vaccination programs.. According to this study, children whose mothers attended a health facility over the previous 12 months were more likely to be fully or partially vaccinated against polio than children whose mothers did not visit a health facility. This result was consistent with studies conducted in sub-Saharan Africa [30] and Pakistan [1]. This might be the fact that mothers who visit health facilities are more likely to obtain information about routine immunization programs and build trust with child healthcare providers.

The study's findings showed that mothers who received a lot of ANC visits had a higher chance of having their children vaccinated against polio. This result is in line with earlier research conducted in Ethiopia [5,31,32], sub-Saharan Africa [30] and, India [29]. Possible reasons for this statement could be that the mothers in Lesotho who visited the ANC contacted health professionals for child health protection and to increase their knowledge about polio vaccination, which helps to reduce healthcare barriers.

This study also revealed that children in Lesotho born to mothers who gave birth in medical facilities were more likely to be partially or completely vaccinated against polio than children born to women who gave birth at home. Research conducted in Kenya [33], Ethiopia [5,31], and India [29] supported this claim. This could be as a result of women who gave birth at medical facilities in Lesotho getting positive influence from trained maternal and pediatric health experts, which raises or prioritizes child polio vaccination. The study's findings also showed that children's propensity to receive vaccinations declined with family size. This explanation was supported by research in Ethiopia [34], Kenya [35], and Africa [36]. This statement's most likely explanation is that a large family size may result in a lack of funds for children's health safeguards, and the parents in Lesotho may have a small number of interactions with health professionals.

Finally, the marginal effects of the independent variables on a particular polio vaccination category were also investigated in this study. According to the final findings, the factors significantly related to child polio vaccination status in Lesotho were family size, number of ANC visits above four, visiting a health facility over the previous 12 months, district (Maseru and Qacha's Nek), and birth at a health facility.

## Strength and limitations

The study's primary strength was the high response rate of the DHS data utilized to analyze the status of polio vaccinations. The study also showed the marginal impacts of each predictor on a particular vaccination category and took into account the cluster-level variability of polio vaccination in Lesotho. Recall bias and the small number of studies in Lesotho pertaining to polio vaccine were considered as study limitations because the study was based on survey DHS data.

## Conclusions

In Lesotho and around the world, polio vaccination is a safe and economical health measure to prevent polio. The main goal of this study was to identify the key factors and marginal effects of polio vaccination status in children between the ages of 12 and 23 months.

Lesotho continues to have a high frequency of children who are polio unvaccinated, according to the findings. Children of mothers with a high number of ANC contacts and those born in medical institutions were more likely to receive a polio

vaccination. In Lesotho, there are notable differences in the polio vaccination status of children between clusters, and the random intercept proportional odds model was used to fit the data. There were significant relationships between polio vaccination status and the variables like district, number of ANC visits, delivery place, family size, and visit to a health institution during the previous 12 months.

Increasing maternal knowledge and fortifying healthcare protections are the primary ways to raise the child polio vaccination rate. Independent estimates of clusters in Lesotho should be used to undertake additional study on polio vaccination.

## Acknowledgments

We would also like to thank the DHS program for providing us the Lesotho demographic and health survey data.

## Author contributions

**Conceptualization:** Kassahun Animut Metkie, Yordanos Sisay Asgedom, Mesfin Abebe, Amanuel Yosef Gebrekidan, Tsion Mulat Tebeje.

**Data curation:** Kassahun Animut Metkie.

**Formal analysis:** Kassahun Animut Metkie.

**Investigation:** Kassahun Animut Metkie.

**Methodology:** Kassahun Animut Metkie.

**Project administration:** Kassahun Animut Metkie, Amanuel Yosef Gebrekidan.

**Resources:** Kassahun Animut Metkie.

**Software:** Kassahun Animut Metkie, Yordanos Sisay Asgedom, Tsion Mulat Tebeje.

**Supervision:** Kassahun Animut Metkie, Seblewongel Gebretsadik Sertsewold, Mesfin Abebe.

**Validation:** Kassahun Animut Metkie, Seblewongel Gebretsadik Sertsewold.

**Visualization:** Kassahun Animut Metkie.

**Writing – original draft:** Kassahun Animut Metkie.

**Writing – review & editing:** Kassahun Animut Metkie, Yordanos Sisay Asgedom, Seblewongel Gebretsadik Sertsewold, Mesfin Abebe, Amanuel Yosef Gebrekidan, Tsion Mulat Tebeje.

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
