## [Editor Report · Decision Letter 0]

21 Jan 2025

PONE-D-25-01328Polio vaccination and marginal effects among children aged 12-23 months in Lesotho using 2024 LDHS data: A multilevel analysis approachPLOS ONE

Dear Dr. Metkie,

Thank you for submitting your manuscript to PLOS ONE. After careful consideration, we feel that it has merit but does not fully meet PLOS ONE’s publication criteria as it currently stands. Therefore, we invite you to submit a revised version of the manuscript that addresses the points raised during the review process.

**See attached comments**==============================

We look forward to receiving your revised manuscript.

Kind regards,

Talkmore Maruta, PhD

Academic Editor

PLOS ONE

2. In the online submission form, you indicated that [The data will available by the corresponding author/DHS program website upon request.].

---

## [Author Response · Author response to Decision Letter 1]

30 Jan 2025

Thank you for your comments for our manuscript.

We revised the manuscript as per your comments and suggestions.

---

## [Editor Report · Decision Letter 1]

5 Feb 2025

PONE-D-25-01328R1Polio vaccination and marginal effects among children aged 12-23 months in Lesotho using 2024 LDHS data: A multilevel analysis approachPLOS ONE

Dear Dr.  Metkie, 

Thank you for submitting your manuscript to PLOS ONE. After careful consideration, we feel that it has merit but does not fully meet PLOS ONE’s publication criteria as it currently stands. Therefore, we invite you to submit a revised version of the manuscript that addresses the points raised during the review process.

Please submit your revised manuscript by Mar 22 2025 11:59PM.  If you will need more time than this to complete your revisions, please reply to this message or contact the journal office at plosone@plos.org . Please include the following items when submitting your revised manuscript:

We look forward to receiving your revised manuscript.

Kind regards,

Talkmore Maruta, PhD

Academic Editor

PLOS ONE

Journal Requirements:

Additional Editor Comments:

**See attached comments**

---

## [Author Response · Author response to Decision Letter 2]

11 Feb 2025

Date: 11 Feb 2025

Response Letter to the Editor and reviewer (s) Comments

JOURNAL: PLOS ONE

Respected Editor and reviewer(s)!

We would like to forward a great gratitude to you for your professional review and scientifically sounded comments used as a corner stone for improving our manuscript entitled “Polio vaccination and marginal effects among children aged 12-23 months in Lesotho using 2024 LDHS data: A multilevel analysis approach"

We have always a great appreciation for the given comments and we accepted your suggestions and comments then revised the manuscript accordingly as described below.

I. Response to the comments about Journal requirements

Response: Dear(s), thank you for the comment and suggestion about the PLOS ONE Journals requirements!

All the references are checked for their correctness/completeness and we didn’t cite any retracted papers in the manuscript.

II. Response to the comments from the attached manuscript

1. About DHS data description…in methods section

This is still a description of another study, and not this study.

Response: Thanks for your constructive comments once again.

- Based on your given comment, we revised this section in the manuscript. See page 4-5 in the manuscript

2. About sample size of 529…….

How was the 529 derived - till not described/explained

Why not 600….or not 800..or not 100………….how did you arrive at 529

Response: Thank you very much dear editor!

We really appreciate for the comments and the manuscript was revised (we added the flow chart that showed how the sample of 529 children was derived from Kids file in LDHS data).

---

## [Editor Report · Decision Letter 2]

27 Mar 2025

Polio vaccination and marginal effects among children aged 12-23 months in Lesotho using 2024 LDHS data: A multilevel analysis approach

PONE-D-25-01328R2

Dear Dr. Metkie,

We’re pleased to inform you that your manuscript has been judged scientifically suitable for publication and will be formally accepted for publication once it meets all outstanding technical requirements.

Kind regards,

Wen-Jun Tu

Academic Editor

PLOS ONE
---

## [Editor Report · Acceptance letter]

PONE-D-25-01328R2

PLOS ONE

Dear Dr. Metkie,

I'm pleased to inform you that your manuscript has been deemed suitable for publication in PLOS ONE. Congratulations! Your manuscript is now being handed over to our production team.

Kind regards,

on behalf of

Dr. Wen-Jun Tu

Academic Editor

PLOS ONE